# Learning to Communicate with Deep Multi-Agent Reinforcement Learning

**Jakob N. Foerster**[1,†]
jakob.foerster@cs.ox.ac.uk

**Yannis M. Assael**[1,†]
yannis.assael@cs.ox.ac.uk

**Nando de Freitas**[1,2,3]
nandodefreitas@google.com

**Shimon Whiteson**[1]
shimon.whiteson@cs.ox.ac.uk

[1]University of Oxford, United Kingdom
[2]Canadian Institute for Advanced Research, CIFAR NCAP Program
[3]Google DeepMind

## Abstract

We consider the problem of multiple agents sensing and acting in environments with the goal of maximising their shared utility. In these environments, agents must learn communication protocols in order to share information that is needed to solve the tasks. By embracing deep neural networks, we are able to demonstrate end-to-end learning of protocols in complex environments inspired by communication riddles and multi-agent computer vision problems with partial observability. We propose two approaches for learning in these domains: Reinforced Inter-Agent Learning (RIAL) and Differentiable Inter-Agent Learning (DIAL). The former uses deep Q-learning, while the latter exploits the fact that, during learning, agents can backpropagate error derivatives through (noisy) communication channels. Hence, this approach uses centralised learning but decentralised execution. Our experiments introduce new environments for studying the learning of communication protocols and present a set of engineering innovations that are essential for success in these domains.

## 1 Introduction

How language and communication emerge among intelligent agents has long been a topic of intense debate. Among the many unresolved questions are: Why does language use discrete structures? What role does the environment play? What is innate and what is learned? And so on. Some of the debates on these questions have been so fiery that in 1866 the French Academy of Sciences banned publications about the origin of human language.

The rapid progress in recent years of machine learning, and deep learning in particular, opens the door to a new perspective on this debate. How can agents use machine learning to automatically discover the communication protocols they need to coordinate their behaviour? What, if anything, can deep learning offer to such agents? What insights can we glean from the success or failure of agents that learn to communicate?

In this paper, we take the first steps towards answering these questions. Our approach is programmatic: first, we propose a set of multi-agent benchmark tasks that require communication; then, we formulate several learning algorithms for these tasks; finally, we analyse how these algorithms learn, or fail to learn, communication protocols for the agents.

---

[†]These authors contributed equally to this work.

The tasks that we consider are fully cooperative, partially observable, sequential multi-agent decision making problems. All the agents share the goal of maximising the same discounted sum of rewards. While no agent can observe the underlying Markov state, each agent receives a private observation correlated with that state. In addition to taking actions that affect the environment, each agent can also communicate with its fellow agents via a discrete limited-bandwidth channel. Due to the partial observability and limited channel capacity, the agents must discover a communication protocol that enables them to coordinate their behaviour and solve the task.

We focus on settings with *centralised learning* but *decentralised execution*. In other words, communication between agents is not restricted during learning, which is performed by a centralised algorithm; however, during execution of the learned policies, the agents can communicate only via the limited-bandwidth channel. While not all real-world problems can be solved in this way, a great many can, e.g., when training a group of robots on a simulator. Centralised planning and decentralised execution is also a standard paradigm for multi-agent planning [1, 2].

To address this setting, we formulate two approaches. The first, *reinforced inter-agent learning* (RIAL), uses deep $Q$-learning [3] with a recurrent network to address partial observability. In one variant of this approach, which we refer to as *independent Q-learning*, the agents each learn their own network parameters, treating the other agents as part of the environment. Another variant trains a single network whose parameters are shared among all agents. Execution remains decentralised, at which point they receive different observations leading to different behaviour.

The second approach, *differentiable inter-agent learning* (DIAL), is based on the insight that centralised learning affords more opportunities to improve learning than just parameter sharing. In particular, while RIAL is end-to-end trainable *within* an agent, it is not end-to-end trainable *across* agents, i.e., no gradients are passed between agents. The second approach allows real-valued messages to pass between agents during centralised learning, thereby treating communication actions as bottleneck connections between agents. As a result, gradients can be pushed through the communication channel, yielding a system that is end-to-end trainable even across agents. During decentralised execution, real-valued messages are discretised and mapped to the discrete set of communication actions allowed by the task. Because DIAL passes gradients from agent to agent, it is an inherently deep learning approach.

Experiments on two benchmark tasks, based on the MNIST dataset and a well known riddle, show, not only can these methods solve these tasks, they often discover elegant communication protocols along the way. To our knowledge, this is the first time that either differentiable communication or reinforcement learning with deep neural networks has succeeded in learning communication protocols in complex environments involving sequences and raw images. The results also show that deep learning, by better exploiting the opportunities of centralised learning, is a uniquely powerful tool for learning such protocols. Finally, this study advances several engineering innovations that are essential for learning communication protocols in our proposed benchmarks.

## 2 Related Work

Research on communication spans many fields, e.g. linguistics, psychology, evolution and AI. In AI, it is split along a few axes: a) predefined or learned communication protocols, b) planning or learning methods, c) evolution or RL, and d) cooperative or competitive settings.

Given the topic of our paper, we focus on related work that deals with the cooperative learning of communication protocols. Out of the plethora of work on multi-agent RL with communication, e.g., [4–7], only a few fall into this category. Most assume a pre-defined communication protocol, rather than trying to learn protocols. One exception is the work of Kasai et al. [7], in which tabular Q-learning agents have to learn the content of a message to solve a predator-prey task with communication. Another example of open-ended communication learning in a multi-agent task is given in [8]. Here evolutionary methods are used for learning the protocols which are evaluated on a similar predator-prey task. Their approach uses a fitness function that is carefully designed to accelerate learning. In general, heuristics and handcrafted rules have prevailed widely in this line of research. Moreover, typical tasks have been necessarily small so that global optimisation methods, such as evolutionary algorithms, can be applied. The use of deep representations and gradient-based optimisation as advocated in this paper is an important departure, essential for scalability and further

progress. A similar rationale is provided in [9], another example of making an RL problem end-to-end differentiable.

Unlike the recent work in [10], we consider discrete communication channels. One of the key components of our methods is the signal binarisation during the decentralised execution. This is related to recent research on fitting neural networks in low-powered devices with memory and computational limitations using binary weights, e.g. [11], and previous works on discovering binary codes for documents [12].

## 3 Background

**Deep Q-Networks (DQN).** In a single-agent, fully-observable, RL setting [13], an agent observes the current state $s_t \in \mathcal{S}$ at each discrete time step $t$, chooses an action $u_t \in \mathcal{U}$ according to a potentially stochastic policy $\pi$, observes a reward signal $r_t$, and transitions to a new state $s_{t+1}$. Its objective is to maximise an expectation over the discounted return, $R_t = r_t + \gamma r_{t+1} + \gamma^2 r_{t+2} + \cdots$, where $r_t$ is the reward received at time $t$ and $\gamma \in [0, 1]$ is a discount factor. The $Q$-function of a policy $\pi$ is $Q^\pi(s, u) = \mathbb{E}\left[R_t | s_t = s, u_t = u\right]$. The optimal action-value function $Q^*(s, u) = \max_\pi Q^\pi(s, u)$ obeys the Bellman optimality equation $Q^*(s, u) = \mathbb{E}_{s'}\left[r + \gamma \max_{u'} Q^*(s', u') \mid s, u\right]$. Deep $Q$-learning [3] uses neural networks parameterised by $\theta$ to represent $Q(s, u; \theta)$. DQNs are optimised by minimising: $\mathcal{L}_i(\theta_i) = \mathbb{E}_{s,u,r,s'}[(y_i^{DQN} - Q(s, u; \theta_i))^2]$, at each iteration $i$, with target $y_i^{DQN} = r + \gamma \max_{u'} Q(s', u'; \theta_i^-)$. Here, $\theta_i^-$ are the parameters of a target network that is frozen for a number of iterations while updating the online network $Q(s, u; \theta_i)$. The action $u$ is chosen from $Q(s, u; \theta_i)$ by an *action selector*, which typically implements an $\epsilon$-greedy policy that selects the action that maximises the Q-value with a probability of $1 - \epsilon$ and chooses randomly with a probability of $\epsilon$. DQN also uses *experience replay*: during learning, the agent builds a dataset of episodic experiences and is then trained by sampling mini-batches of experiences.

**Independent DQN.** DQN has been extended to cooperative multi-agent settings, in which each agent $a$ observes the global $s_t$, selects an individual action $u_t^a$, and receives a team reward, $r_t$, shared among all agents. Tampuu et al. [14] address this setting with a framework that combines DQN with *independent Q-learning*, in which each agent $a$ independently and simultaneously learns its own Q-function $Q^a(s, u^a; \theta_i^a)$. While independent Q-learning can in principle lead to convergence problems (since one agent's learning makes the environment appear non-stationary to other agents), it has a strong empirical track record [15, 16], and was successfully applied to two-player pong.

**Deep Recurrent Q-Networks.** Both DQN and independent DQN assume full observability, i.e., the agent receives $s_t$ as input. By contrast, in partially observable environments, $s_t$ is hidden and the agent receives only an observation $o_t$ that is correlated with $s_t$, but in general does not disambiguate it. Hausknecht and Stone [17] propose *deep recurrent Q-networks* (DRQN) to address single-agent, partially observable settings. Instead of approximating $Q(s, u)$ with a feed-forward network, they approximate $Q(o, u)$ with a recurrent neural network that can maintain an internal state and aggregate observations over time. This can be modelled by adding an extra input $h_{t-1}$ that represents the hidden state of the network, yielding $Q(o_t, h_{t-1}, u)$. For notational simplicity, we omit the dependence of $Q$ on $\theta$.

## 4 Setting

In this work, we consider RL problems with both multiple agents and partial observability. All the agents share the goal of maximising the same discounted sum of rewards $R_t$. While no agent can observe the underlying Markov state $s_t$, each agent $a$ receives a private observation $o_t^a$ correlated with $s_t$. In every time-step $t$, each agent selects an *environment action* $u_t^a \in U$ that affects the environment, and a *communication action* $m_t^a \in M$ that is observed by other agents but has no direct impact on the environment or reward. We are interested in such settings because it is only when multiple agents and partial observability coexist that agents have the incentive to communicate. As no communication protocol is given a priori, the agents must develop and agree upon such a protocol to solve the task.

Since protocols are mappings from action-observation histories to sequences of messages, the space of protocols is extremely high-dimensional. Automatically discovering effective protocols in this space remains an elusive challenge. In particular, the difficulty of exploring this space of protocols is exacerbated by the need for agents to coordinate the sending and interpreting of messages. For

example, if one agent sends a useful message to another agent, it will only receive a positive reward if the receiving agent correctly interprets and acts upon that message. If it does not, the sender will be discouraged from sending that message again. Hence, positive rewards are sparse, arising only when sending and interpreting are properly coordinated, which is hard to discover via random exploration.

We focus on settings where communication between agents is not restricted during *centralised learning*, but during the *decentralised execution* of the learned policies, the agents can communicate only via a limited-bandwidth channel.

# 5 Methods

In this section, we present two approaches for learning communication protocols.

## 5.1 Reinforced Inter-Agent Learning

The most straightforward approach, which we call *reinforced inter-agent learning* (RIAL), is to combine DRQN with independent Q-learning for action and communication selection. Each agent's Q-network represents $Q^a(o_t^a, m_{t-1}^{a'}, h_{t-1}^a, u^a)$, which conditions on that agent's individual hidden state $h_{t-1}^a$ and observation $o_t^a$ as well as messages from other agents $m_{t-1}^{a'}$.

To avoid needing a network with $|U||M|$ outputs, we split the network into $Q_u^a$ and $Q_m^a$, the Q-values for the environment and communication actions, respectively. Similarly to [18], the action selector separately picks $u_t^a$ and $m_t^a$ from $Q_u$ and $Q_m$, using an $\epsilon$-greedy policy. Hence, the network requires only $|U| + |M|$ outputs and action selection requires maximising over $U$ and then over $M$, but not maximising over $U \times M$.

Both $Q_u$ and $Q_m$ are trained using DQN with the following two modifications, which were found to be essential for performance. First, we disable experience replay to account for the non-stationarity that occurs when multiple agents learn concurrently, as it can render experience obsolete and misleading. Second, to account for partial observability, we feed in the actions $u$ and $m$ taken by each agent as inputs on the next time-step. Figure 1(a) shows how information flows between agents and the environment, and how Q-values are processed by the action selector in order to produce the action, $u_t^a$, and message $m_t^a$. Since this approach treats agents as independent networks, the learning phase is not centralised, even though our problem setting allows it to be. Consequently, the agents are treated exactly the same way during decentralised execution as during learning.

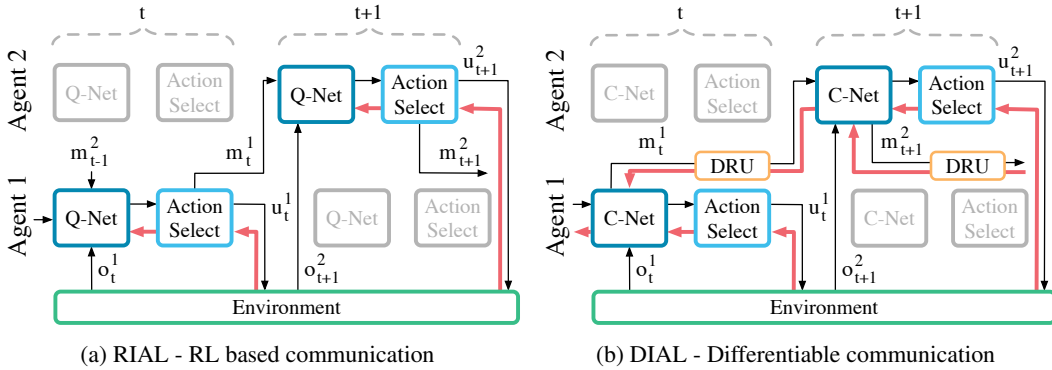

(a) RIAL - RL based communication        (b) DIAL - Differentiable communication

Figure 1: The bottom and top rows represent the communication flow for agent $a_1$ and agent $a_2$, respectively. In RIAL (a), all Q-values are fed to the action selector, which selects both environment and communication actions. Gradients, shown in red, are computed using DQN for the selected action and flow only through the Q-network of a single agent. In DIAL (b), the message $m_t^a$ bypasses the action selector and instead is processed by the DRU (Section 5.2) and passed as a continuous value to the next C-network. Hence, gradients flow across agents, from the recipient to the sender. For simplicity, at each time step only one agent is highlighted, while the other agent is greyed out.

**Parameter Sharing.** RIAL can be extended to take advantage of the opportunity for centralised learning by sharing parameters among the agents. This variation learns only one network, which is used by all agents. However, the agents can still behave differently because they receive different

observations and thus evolve different hidden states. In addition, each agent receives its own index $a$ as input, allowing it to specialise. The rich representations in deep Q-networks can facilitate the learning of a common policy while also allowing for specialisation. Parameter sharing also dramatically reduces the number of parameters that must be learned, thereby speeding learning. Under parameter sharing, the agents learn two $Q$-functions $Q_u(o_t^a, m_{t-1}^{a'}, h_{t-1}^a, u_{t-1}^a, m_{t-1}^a, a, u_t^a)$ and $Q_m(o_t^a, m_{t-1}^{a'}, h_{t-1}^a, u_{t-1}^a, m_{t-1}^a, a, u_t^a)$. During decentralised execution, each agent uses its own copy of the learned network, evolving its own hidden state, selecting its own actions, and communicating with other agents only through the communication channel.

## 5.2 Differentiable Inter-Agent Learning

While RIAL can share parameters among agents, it still does not take full advantage of centralised learning. In particular, the agents do not give each other feedback about their communication actions. Contrast this with human communication, which is rich with tight feedback loops. For example, during face-to-face interaction, listeners send fast nonverbal queues to the speaker indicating the level of understanding and interest. RIAL lacks this feedback mechanism, which is intuitively important for learning communication protocols.

To address this limitation, we propose *differentiable inter-agent learning* (DIAL). The main insight behind DIAL is that the combination of centralised learning and Q-networks makes it possible, not only to share parameters but to push gradients from one agent to another through the communication channel. Thus, while RIAL is end-to-end trainable *within* each agent, DIAL is end-to-end trainable *across* agents. Letting gradients flow from one agent to another gives them richer feedback, reducing the required amount of learning by trial and error, and easing the discovery of effective protocols.

DIAL works as follows: during centralised learning, communication actions are replaced with direct connections between the output of one agent's network and the input of another's. Thus, while the task restricts communication to discrete messages, during learning the agents are free to send real-valued messages to each other. Since these messages function as any other network activation, gradients can be passed back along the channel, allowing end-to-end backpropagation across agents.

In particular, the network, which we call a C-Net, outputs two distinct types of values, as shown in Figure 1(b), a) $Q(\cdot)$, the Q-values for the environment actions, which are fed to the action selector, and b) $m_t^a$, the real-valued vector message to other agents, which bypasses the action selector and is instead processed by the *discretise/regularise unit* (DRU($m_t^a$)). The DRU regularises it during centralised learning, DRU($m_t^a$) = Logistic($\mathcal{N}(m_t^a, \sigma)$), where $\sigma$ is the standard deviation of the noise added to the channel, and discretises it during decentralised execution, DRU($m_t^a$) = $\mathbb{1}\{m_t^a > 0\}$. Figure 1 shows how gradients flow differently in RIAL and DIAL. The gradient chains for $Q_u$, in RIAL and $Q$, in DIAL, are based on the DQN loss. However, in DIAL the gradient term for $m$ is the backpropagated error from the recipient of the message to the sender. Using this inter-agent gradient for training provides a richer training signal than the DQN loss for $Q_m$ in RIAL. While the DQN error is nonzero only for the selected message, the incoming gradient is a $|m|$-dimensional vector that can contain more information. It also allows the network to directly adjust messages in order to minimise the downstream DQN loss, reducing the need for trial and error learning of good protocols.

While we limit our analysis to discrete messages, DIAL naturally handles continuous message spaces, as they are used anyway during centralised learning. At the same time, DIAL can also scale to large discrete message spaces, since it learns binary encodings instead of the one-hot encoding in RIAL, $|m| = O(\log(|M|))$. Further algorithmic details and pseudocode are in the supplementary material.

## 6 Experiments

In this section, we evaluate RIAL and DIAL with and without parameter sharing in two multi-agent problems and compare it with a no-communication shared-parameter baseline (NoComm). Results presented are the average performance across several runs, where those without parameter sharing (-NS), are represented by dashed lines. Across plots, rewards are normalised by the highest average reward achievable given access to the true state (Oracle). In our experiments, we use an $\epsilon$-greedy policy with $\epsilon = 0.05$, the discount factor is $\gamma = 1$, and the target network is reset every 100 episodes. To stabilise learning, we execute parallel episodes in batches of 32. The parameters are optimised using RMSProp [19] with a learning rate of $5 \times 10^{-4}$. The architecture uses *rectified linear units*

(ReLU), and *gated recurrent units* (GRU) [20], which have similar performance to *long short-term memory* [21] (LSTM) [22]. Unless stated otherwise, we set the standard deviation of noise added to the channel to $\sigma = 2$, which was found to be essential for good performance.[1]

## 6.1 Model Architecture

RIAL and DIAL share the same individual model architecture. For brevity, we describe only the DIAL model here. As illustrated in Figure 2, each agent consists of a recurrent neural network (RNN), unrolled for $T$ time-steps, that maintains an internal state $h$, an input network for producing a task embedding $z$, and an output network for the $Q$-values and the messages $m$. The input for agent $a$ is defined as a tuple of $(o_t^a, m_{t-1}^{a'}, u_{t-1}^a, a)$. The inputs $a$ and $u_{t-1}^a$ are passed through lookup tables, and $m_{t-1}^{a'}$ through a 1-layer MLP, both producing embeddings of size 128. $o_t^a$ is processed through a task-specific network that produces an additional embedding of the same size. The state embedding is produced by element-wise summation of

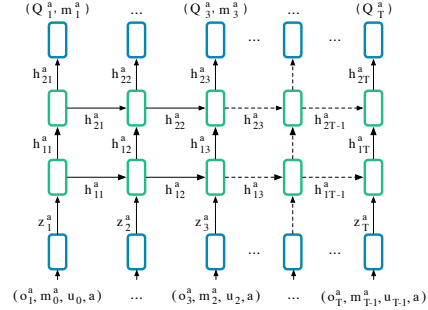

Figure 2: DIAL architecture.

these embeddings, $z_t^a = \left( \text{TaskMLP}(o_t^a) + \text{MLP}[|M|, 128](m_{t-1}) + \text{Lookup}(u_{t-1}^a) + \text{Lookup}(a) \right)$. We found that performance and stability improved when a batch normalisation layer [23] was used to preprocess $m_{t-1}$. $z_t^a$ is processed through a 2-layer RNN with GRUs, $h_{1,t}^a = \text{GRU}[128, 128](z_t^a, h_{1,t-1}^a)$, which is used to approximate the agent's action-observation history. Finally, the output $h_{2,t}^a$ of the top GRU layer, is passed through a 2-layer MLP $Q_t^a, m_t^a = \text{MLP}[128, 128, (|U| + |M|)](h_{2,t}^a)$.

## 6.2 Switch Riddle

The first task is inspired by a well-known riddle described as follows: *"One hundred prisoners have been newly ushered into prison. The warden tells them that starting tomorrow, each of them will be placed in an isolated cell, unable to communicate amongst each other. Each day, the warden will choose one of the prisoners uniformly at random with replacement, and place him in a central interrogation room containing only a light bulb with a toggle switch. The prisoner will be able to observe the current state of the light bulb. If he wishes, he can toggle the light bulb. He also has the option of announcing that he believes all prisoners have visited the interrogation room at some point in time. If this announcement is true, then all prisoners are set free, but if it is false, all prisoners are executed[...]"* [24].

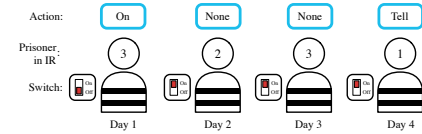

Figure 3: *Switch:* Every day one prisoner gets sent to the interrogation room where he sees the switch and chooses from "On", "Off", "Tell" and "None".

**Architecture.** In our formalisation, at time-step $t$, agent $a$ observes $o_t^a \in \{0, 1\}$, which indicates if the agent is in the interrogation room. Since the switch has two positions, it can be modelled as a 1-bit message, $m_t^a$. If agent $a$ is in the interrogation room, then its actions are $u_t^a \in \{\text{"None"}, \text{"Tell"}\}$; otherwise the only action is "None". The episode ends when an agent chooses "Tell" or when the maximum time-step, $T$, is reached. The reward $r_t$ is 0 unless an agent chooses "Tell", in which case it is 1 if all agents have been to the interrogation room and $-1$ otherwise. Following the riddle definition, in this experiment $m_{t-1}^a$ is available only to the agent $a$ in the interrogation room. Finally, we set the time horizon $T = 4n - 6$ in order to keep the experiments computationally tractable.

**Complexity.** The switch riddle poses significant protocol learning challenges. At any time-step $t$, there are $|o|^t$ possible observation histories for a given agent, with $|o| = 3$: the agent either is not in the interrogation room or receives one of two messages when it is. For each of these histories, an agent can chose between $4 = |U||M|$ different options, so at time-step $t$, the single-agent policy space is $(|U||M|)^{|o|^t} = 4^{3^t}$. The product of all policies for all time-steps defines the total policy space for an agent: $\prod 4^{3^t} = 4^{(3^{T+1}-3)/2}$, where $T$ is the final time-step. The size of the multi-agent

[1]Source code is available at: `https://github.com/iassael/learning-to-communicate`

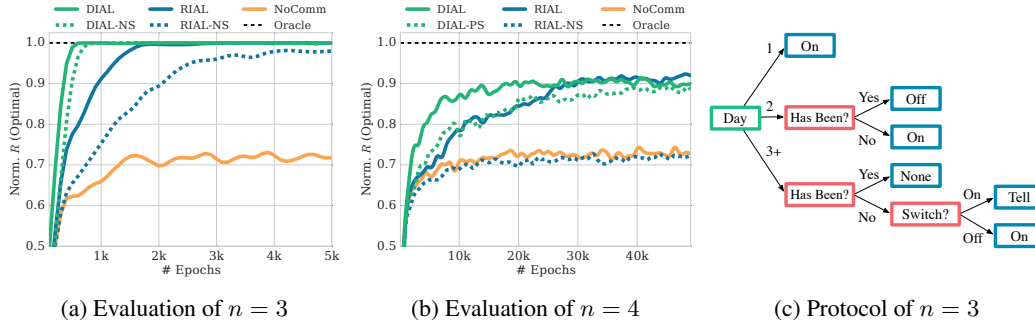

(a) Evaluation of $n = 3$      (b) Evaluation of $n = 4$      (c) Protocol of $n = 3$

Figure 4: *Switch:* (a-b) Performance of DIAL and RIAL, with and without ( -NS) parameter sharing, and NoComm-baseline, for $n = 3$ and $n = 4$ agents. (c) The decision tree extracted for $n = 3$ to interpret the communication protocol discovered by DIAL.

policy space grows exponentially in $n$, the number of agents: $4^{n(3^{T+1}-3)/2}$. We consider a setting where $T$ is proportional to the number of agents, so the total policy space is $4^{n3^{O(n)}}$. For $n = 4$, the size is $4^{354288}$. Our approach using DIAL is to model the switch as a continuous message, which is binarised during decentralised execution.

**Experimental results.** Figure 4(a) shows our results for $n = 3$ agents. All four methods learn an optimal policy in 5k episodes, substantially outperforming the NoComm baseline. DIAL with parameter sharing reaches optimal performance substantially faster than RIAL. Furthermore, parameter sharing speeds both methods. Figure 4(b) shows results for $n = 4$ agents. DIAL with parameter sharing again outperforms all other methods. In this setting, RIAL without parameter sharing was unable to beat the NoComm baseline. These results illustrate how difficult it is for agents to learn the same protocol independently. Hence, parameter sharing can be crucial for learning to communicate. DIAL-NS performs similarly to RIAL, indicating that the gradient provides a richer and more robust source of information. We also analysed the communication protocol discovered by DIAL for $n = 3$ by sampling 1K episodes, for which Figure 4(c) shows a decision tree corresponding to an optimal strategy. When a prisoner visits the interrogation room after day two, there are only two options: either one or two prisoners may have visited the room before. If three prisoners had been, the third prisoner would have finished the game. The other options can be encoded via the "On" and "Off" positions respectively.

## 6.3 MNIST Games

In this section, we consider two tasks based on the well known MNIST digit classification dataset [25].

**Colour-Digit MNIST** is a two-player game in which each agent observes the pixel values of a random MNIST digit in red or green, while the colour label and digit value are hidden. The reward consists of two components that are antisymmetric in the action, colour, and parity of the digits. As only one bit of information can be sent, agents must agree to encode/decode either colour or parity, with parity yielding greater rewards. The game has two steps;

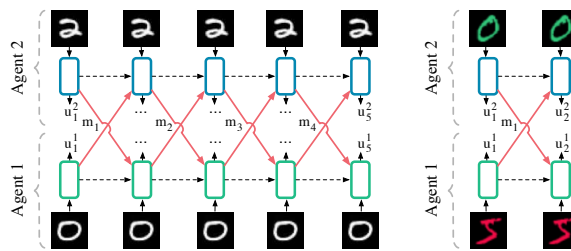

Figure 5: MNIST games architectures.

in the first step, both agents send a 1-bit message, in the second step they select a binary action.

**Multi-Step MNIST** is a grayscale variant that requires agents to develop a communication protocol that integrates information across 5 time-steps in order to guess each others' digits. At each step, the agents exchange a 1-bit message and at he final step, $t = 5$, they are awarded $r = 0.5$ for each correctly guessed digit. Further details on both tasks are provided in the supplementary material.

**Architecture.** The input processing network is a 2-layer MLP TaskMLP$[(|c| \times 28 \times 28), 128, 128](o_t^a)$. Figure 5 depicts the generalised setting for both games. Our experimental evaluation showed improved training time using batch normalisation after the first layer.

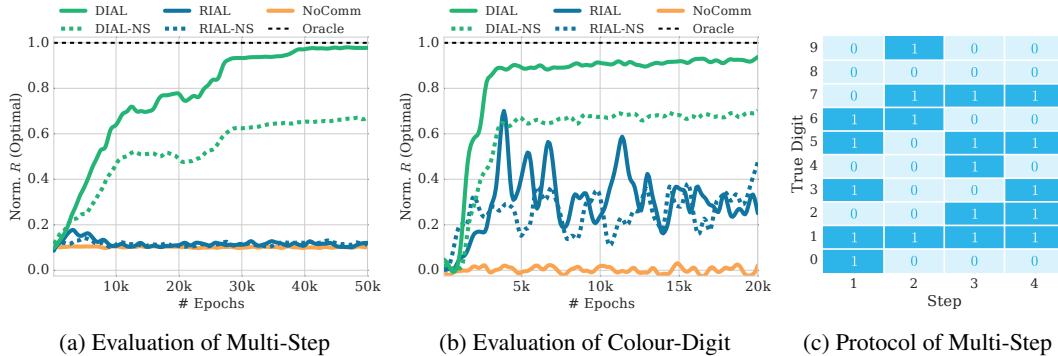

| | (a) Evaluation of Multi-Step | (b) Evaluation of Colour-Digit | (c) Protocol of Multi-Step |

Figure 6: *MNIST Games:* (a,b) Performance of DIAL and RIAL, with and without (-NS) parameter sharing, and NoComm, for both MNIST games. (c) Extracted coding scheme for multi-step MNIST.

**Experimental results.** Figures 6(a) and 6(b) show that DIAL substantially outperforms the other methods on both games. Furthermore, parameter sharing is crucial for reaching the optimal protocol. In multi-step MNIST, results were obtained with $\sigma = 0.5$. In this task, RIAL fails to learn, while in colour-digit MNIST it fluctuates around local minima in the protocol space; the NoComm baseline is stagnant at zero. DIAL's performance can be attributed to directly optimising the messages in order to reduce the global DQN error while RIAL must rely on trial and error. DIAL can also optimise the message content with respect to rewards taking place many time-steps later, due to the gradient passing between agents, leading to optimal performance in multi-step MNIST. To analyse the protocol that DIAL learned, we sampled 1K episodes. Figure 6(c) illustrates the communication bit sent at time-step $t$ by agent 1, as a function of its input digit. Thus, each agent has learned a binary encoding and decoding of the digits. These results illustrate that differentiable communication in DIAL is essential to fully exploiting the power of centralised learning and thus is an important tool for studying the learning of communication protocols.

### 6.4 Effect of Channel Noise

The question of why language evolved to be discrete has been studied for centuries, see e.g., the overview in [26]. Since DIAL learns to communicate in a continuous channel, our results offer an illuminating perspective on this topic. In particular, Figure 7 shows that, in the switch riddle, DIAL without noise in the communication channel learns centred activations. By contrast, the presence of noise forces messages into two different modes during learning. Similar observations have been made in relation to adding noise when training document models [12] and performing classification [11]. In our work, we found that adding noise was essential for successful training. More analysis on this appears in the supplementary material.

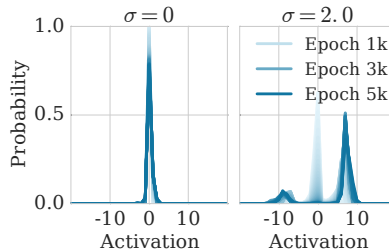

Figure 7: DIAL's learned activations with and without noise in DRU.

## 7 Conclusions

This paper advanced novel environments and successful techniques for learning communication protocols. It presented a detailed comparative analysis covering important factors involved in the learning of communication protocols with deep networks, including differentiable communication, neural network architecture design, channel noise, tied parameters, and other methodological aspects.

This paper should be seen as a first attempt at learning communication and language with deep learning approaches. The gargantuan task of understanding communication and language in their full splendour, covering compositionality, concept lifting, conversational agents, and many other important problems still lies ahead. We are however optimistic that the approaches proposed in this paper can help tackle these challenges.

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
