[Supplementary Material · paper_supplementary.pdf]

# Learning to Communicate with
# Deep Multi-Agent Reinforcement Learning
# Supplementary Material

**Jakob N. Foerster**[1,†]
jakob.foerster@cs.ox.ac.uk

**Yannis M. Assael**[1,†]
yannis.assael@cs.ox.ac.uk

**Nando de Freitas**[1,2,3]
nandodefreitas@google.com

**Shimon Whiteson**[1]
shimon.whiteson@cs.ox.ac.uk

[1]University of Oxford, United Kingdom
[2]Canadian Institute for Advanced Research, CIFAR NCAP Program
[3]Google DeepMind

## A    DIAL Details

Algorithm 1 formally describes DIAL. At each time-step, we pick an action for each agent $\epsilon$-greedily with respect to the Q-function and assign an outgoing message

$$Q(\cdot), m_t^a = \text{C-Net}\left(o_t^a, \hat{m}_{t-1}^{a'}, h_{t-1}^a, u_{t-1}^a, a; \theta_i\right). \tag{1}$$

We feed in the previous action, $u_{t-1}^a$, the agent index, $a$, along with the observation $o_t^a$, the previous internal state, $h_{t-1}^a$ and the incoming messages $\hat{m}_{t-1}^{a'}$ from other agents. After all agents have taken their actions, we query the environment for a state update and reward information.

When we reach the final time-step or a terminal state, we proceed to the backwards pass. Here, for each agent, $a$, and time-step, $j$, we calculate a target Q-value, $y_j^a$, using the observed reward, $r_t$, and the discounted target network. We then accumulate the gradients, $\nabla\theta$, by regressing the Q-value estimate

$$Q(o_t^a, \hat{m}_{t-1}^{a'}, h_{t-1}^a, u_{t-1}^a, a, u; \theta_i), \tag{2}$$

against the target Q-value, $y_t^a$, for the chosen action, $u_t^a$. We also update the message gradient chain $\mu_t^a$, which contains the derivative of the downstream bootstrap error $\sum_{m,t'>t}\left(\Delta Q_{t+1}^{a'}\right)^2$ with respect to the outgoing message $m_t^a$.

To allow for efficient calculation, this sum can be broken into two parts. The first part, $\sum_{m'\neq m}\frac{\partial}{\partial \hat{m}_t^a}\left(\Delta Q_{t+1}^{a'}\right)^2$, captures the impact of the message on the total estimation error of the next step. The impact of the message $m_t^a$ on all other future rewards $t' > t + 1$ can be calculated using the partial derivative of the outgoing messages from the agents at time $t + 1$ with respect to the incoming message $m_t^a$, multiplied with their message gradients, $\mu_{t+1}^{a'}$. Using the message gradient, we can calculate the derivative with respect to the parameters, $\mu_t^a\frac{\partial \hat{m}_t^a}{\partial \theta}$.

Having accumulated all gradients, we conduct two parameter updates, first $\theta_i$ in the direction of the accumulated gradients, $\nabla\theta$, and then every C steps $\theta_i^- = \theta_i$. During decentralised execution, the outgoing activations in the channel are mapped into a binary vector, $\hat{m} = \mathbb{1}\{m_t^a > 0\}$. This ensures that discrete messages are exchanged, as required by the task.

---

---

**Algorithm 1** Differentiable Communication (DIAL)

---

Initialise $\theta_1$ and $\theta_1^-$
**for** each episode $e$ **do**
    $s_1$ = initial state, $t = 0$, $h_0^a = \mathbf{0}$ for each agent $a$
    **while** $s_t \neq$ terminal **and** $t < T$ **do**
        $t = t + 1$
        **for** each agent $a$ **do**
            Get messages $\hat{m}_{t-1}^{a'}$ of previous time-steps from agents $m'$ and evaluate C-Net:

$$Q(\cdot), m_t^a = \text{C-Net}\left(o_t^a, \hat{m}_{t-1}^{a'}, h_{t-1}^a, u_{t-1}^a, a; \theta_i\right)$$

            With probability $\epsilon$ pick random $u_t^a$, else $u_t^a = \max_a Q\left(o_t^a, \hat{m}_{t-1}^{a'}, h_{t-1}^a, u_{t-1}^a, a, u; \theta_i\right)$

            Set message $\hat{m}_t^a = \text{DRU}(m)$, where $\text{DRU}(m) = \begin{cases} \text{Logistic}(\mathcal{N}(m, \sigma)), \text{ if training} \\ \mathbb{1}\{m > 0\}, \text{ otherwise} \end{cases}$

        Get reward $r_t$ and next state $s_{t+1}$
    Reset gradients $\nabla\theta = 0$
    **for** $t = T$ **to** $1, -1$ **do**
        **for** each agent $a$ **do**

$$y_t^a = \begin{cases} r_t, \text{ if } s_t \text{ terminal, else} \\ r_t + \gamma\max_u Q\left(o_{t+1}^a, \hat{m}_t^{a'}, h_t^a, u_t^a, a, u; \theta_i^-\right) \end{cases}$$

            Accumulate gradients for action:

$$\Delta Q_t^a = y_t^a - Q\left(o_j^a, h_{t-1}^a, \hat{m}_{t-1}^{a'}, u_{t-1}^a, a, u_t^a; \theta_i\right)$$
$$\nabla\theta = \nabla\theta + \frac{\partial}{\partial\theta}(\Delta Q_t^a)^2$$

            Update gradient chain for differentiable communication:

$$\mu_j^a = \mathbb{1}\{t < T-1\}\sum_{m' \neq m}\frac{\partial}{\partial\hat{m}_t^a}\left(\Delta Q_{t+1}^{a'}\right)^2 + \mu_{t+1}^{a'}\frac{\partial\hat{m}_{t+1}^{a'}}{\partial\hat{m}_t^a}$$

            Accumulate gradients for differentiable communication:

$$\nabla\theta = \nabla\theta + \mu_t^a\frac{\partial}{\partial m_t^a}\text{DRU}(m_t^a)\frac{\partial m_t^a}{\partial\theta}$$

    $\theta_{i+1} = \theta_i + \alpha\nabla\theta$
    Every $C$ steps reset $\theta_i^- = \theta_i$

---

In order to minimise the discretisation error when mapping from continuous values to discrete encodings, two measures are taken during centralised learning. First, Gaussian noise is added in order to limit the number of bits that can be encoded in a given range of $m$ values. Second, the noisy message is passed through a logistic function to restrict the range available for encoding information. Together, these two measures regularise the information transmitted through the bottleneck. Furthermore, the noise also perturbs values in the middle of the range, due to the steeper slope, but leaves the tails of the distribution unchanged.

Formally, during centralised learning, $m$ is mapped to $\hat{m} = \text{Logistic}(\mathcal{N}(m, \sigma))$, where $\sigma$ is chosen to be comparable to the width of the logistic function. In Algorithm 1, the mapping logic from $m$ to $\hat{m}$ during training and execution is contained in the $\text{DRU}(m_t^a)$ function.

## B MNIST Games: Further Analysis

Our results show that DIAL deals more effectively with stochastic rewards in the *colour-digit MNIST* game than RIAL. To better understand why, consider a simpler two-agent problem with a structurally similar reward function $r = (-1)^{(s^1+s^2+a^2)}$, which is antisymmetric in the observations and action of the agents. Here random digits $s^1, s^2 \in 0, 1$ are input to agent 1 and agent 2 and $u^2 \in 1, 2$ is a binary action. Agent 1 can send a single bit message, $m^1$. Until a protocol has been learned, the average reward for any action by agent 2 is 0, since averaged over $s_1$ the reward has an equal probability of being $+1$ or $-1$. Equally the TD error for agent 1, the sender, is zero for any message $m$:

$$\mathbb{E}\left[\Delta Q(s^1, m^1)\right] = Q(s^1, m^1) - \mathbb{E}\left[r(s^2, a^2, s^1)\right]_{s^2, a^2} = 0 - 0, \tag{3}$$

By contrast, DIAL allows for learning. Unlike the TD error, the gradient is a function of the action and the observation of the receiving agent, so summed across different $+1/-1$ outcomes the gradient updates for the message $m$ no longer cancel:

$$\mathbb{E}\left[\nabla\theta\right] = \mathbb{E}\left[\left(Q(s^2, m^1, a^2) - r(s^2, a^2, s^1)\right) \frac{\partial}{\partial m}Q(s^2, m^1, a^2)\frac{\partial}{\partial \theta}m^1(s^1)\right]_{<s^2,a^2>}. \quad (4)$$

## C    Effect of Noise: Further Analysis

Given that the amount of noise, $\sigma$, is a hyperparameter that needs to be set, it is useful to understand how it impacts the amount of information that can pass through the channel. A first intuition can be gained by looking at the width of the sigmoid: Taking the decodable range of the logistic function to be $x$ values corresponding to $y$ values between 0.01 and 0.99, an initial estimate for the range is $\approx 10$. Thus, requiring distinct $x$ values to be at least six standard deviations apart, with $\sigma = 2$, only two bits can be encoded reliably in this range. To get a better understanding of the required $\sigma$ we can visualise the capacity of the channel including the logistic function and the Gaussian noise. To do so, we must first derive an expression for the probability distribution of outgoing messages, $\hat{m}$, given incoming activations, $m$, $P(\hat{m}|m)$:

$$P(\hat{m}|m) = \frac{1}{\sqrt{2\pi}\sigma\hat{m}(1-\hat{m})} \exp\left(-\frac{\left(m - \log(\frac{1}{\hat{m}} - 1)\right)^2}{\sigma^2}\right). \quad (5)$$

For any $m$, this captures the distribution of messages leaving the channel. Two $m$ values $m_1$ and $m_2$ can be distinguished when the outgoing messages have a small probability of overlapping. Given a value $m_1$ we can thus pick a next value $m_2$ to be distinguishable when the highest value $\hat{m}_1$ that $m_1$ is likely to produce is less than the lowest value $\hat{m}_2$ that $m_2$ is likely to produce. An approximation for when this happens is when $(\max_{\hat{m}} s.t.P(\hat{m}|m_1) > \epsilon) = (\min_{\hat{m}} s.t.P(\hat{m}|m_2) > \epsilon)$. Figure 1 illustrates this for three different values of $\sigma$. For $\sigma > 2$, only two options can be reliably encoded using $\epsilon = 0.1$, resulting in a channel that effectively transmits only one bit of information.

Figure 1: Distribution of regularised messages, $P(\hat{m}|m)$ for different noise levels. Shading indicates $P(\hat{m}|m) > 0.1$. Blue bars show a division of the $x$-range into intervals s.t. the resulting $y$-values have a small probability of overlap, leading to decodable values.

Interestingly, the amount of noise required to regularise the channel depends greatly on the benefits of over-encoding information. More specifically, as illustrated in Figure 2, in tasks where sending more bits does not lead to higher rewards, small amounts of noise are sufficient to encourage discretisation, as the network can maximise reward by pushing activations to the tails of the sigmoid, where the noise is minimised. The figure illustrates the final average evaluation performance normalised by the training performance of three runs after 50K of the multi-step MNIST game, under different noise

Figure 2: Final evaluation performance on multi-step MNIST of DIAL normalised by training performance after 50K epochs, under different noise regularisation levels $\sigma \in \{0, 0.5, 1, 1.5, 2\}$, and different numbers of steps $step \in [2, \ldots, 5]$.

regularisation levels $\sigma \in \{0, 0.5, 1, 1.5, 2\}$, and different numbers of steps $step \in [2, \ldots, 5]$. When the lines exceed "Regularised", the test reward, after discretisation, is higher than the training reward, i.e., the channel is properly regularised and getting used as a single bit at the end of learning. Given that there are 10 digits to encode, four bits are required to get full reward. Reducing the number of steps directly reduces the number of bits that can be communicated, $\#bits = steps - 1$, and thus creates an incentive for the network to over-encode information in the channel, which leads to greater discretisation error. This is confirmed by the normalised performance for $\sigma = 0.5$, which is around 0.7 for 2 steps (1 bit) and then goes up to $> 1$ for 5 steps (4 bits). Note also that, without noise, regularisation is not possible and that with enough noise the channel is always regularised, even if over-encoding information would yield higher training rewards.