[Reviews · NeurIPS 2016]

Reviewer 1

Summary

This paper considers the cooperative learning of communication protocols. Two approaches, reinforcement inter-agent learning (RIAL) and differentiable inter-agent learning (DIAL), are proposed for fully cooperative, partially observable, sequential multi-agent decision making problems, with the objective of maximizing a common discounted sum of rewards. In RIAL, two separate Q-networks are used for the environment and communication actions, respectively, and each agent is trained in an end-to-end manner according to the Q-learning objective function. In DIAL, the communication signal and the environment actions are output from a common C-Net, and the entire multi-agent system formed a big neural network so that the entire multi-agent system is trained in an end-to-end manner.

Qualitative Assessment

The paper considers the settings with centralized learning but decentralized execution, where communication between agents is not restricted during learning but is restricted over band-limited channel during execution. An important problem here is whether such a mismatch between training and execution settings will lead to performance degradation. The authors may want to comment on this. Is m_t^a in DIAL a binary vector or a binary scalar?

Confidence in this Review

2-Confident (read it all; understood it all reasonably well)


Reviewer 2

Summary

The paper proposes two novel approaches for multi-agents communication in the context of partially observable problems. The goal is to define methods for learning communication protocols that allow the agents to communicate for collaboration. The problem setting allows here centralized learning. Both approaches are based on reinforcement Deep Recurrent Q-learning Networks (DRQN), which they employ in the context of multi-agents collaboration. However, while the first approach is only a straightforward application of DRQN in the context of multi-agents, the second proposes real-valued communication between agents that allows a back-propagation of the gradient via the communication channel, which favors the difficult emergence of an effective communication protocol between agents.

Qualitative Assessment

Both proposed approaches look useful and relevant, although I am not fully convinced by the great novelty of the presented work, as the proposals appear to me as rather straightforward applications of DRQN to the setting of multi-agents. Experiments, which appear appropriate, demonstrate the good behavior of the approaches. Especially, what looks very interesting to me (and should probably be more highlighted in the results analysis) is the performance of DIAL-NS compared to RIAL-NS. It demonstrates the importance of a feedback on the communication protocol when no parameter sharing is possible w.r.t. the setting. I like the background section and the introduction of DIAL. However, the presentation of the approaches (section 5) should be reworked for a better understanding (I needed to come back to this section several times to well understand the proposals). Some justifications are also missing in the paper. Notably: - Why adding the last action inputs in the Q functions of RIAL with parameter sharing, What is the intuition behind this ? - What is the intuition between using DRU ? Why discretising during execution and regularizing during training ? What justifies the add of a noise in the communication channel ? (ok, the importance of the noise is discussed in 6.4 but from my point of view the intuition should be introduced earlier). Experiments should also consider more massive / real-world evaluation, rather than only toy data/tasks such as what is considered in the paper. Minor issues: - the last paragraph of section 4 is the same as the first of page 2 - the 2 first sentences of the last paragraph of 5.2 are the same - m^{a'}_{t-1} is not defined when firstly used line 149. - line 226: what is sigma ? the standard deviation defined line 200 ? - Performance measures used in experiments are not described - The description of the MNIST Games should be reworked for more clarity

Confidence in this Review

2-Confident (read it all; understood it all reasonably well)


Reviewer 3

Summary

This paper proposed a deep learning based method for learning communication protocols in cooperative multi-agent systems. Agents need to maximize their shared utility through interacting with the partially observable environment.

Qualitative Assessment

The proposed idea is novel and the paper is well-organized. There are some slight issues. DRQN is not explained before using (in Section 5.1). what does the a' in Section 5.1 stand for? A single agent or a set of agent? I assume it's a set of agent because the model support #agents greater than 2.

Confidence in this Review

1-Less confident (might not have understood significant parts)


Reviewer 4

Summary

The authors present different variants of an algorithm that solves multi-agent reinforcement learning problems where planning is central but execution is autonomous. The main contribution is a network architecture that allows inter-agent communication in partial observable scenarios. The authors show that the agents are able to learn complex communication protocols in challenging tasks.

Qualitative Assessment

The authors show that their extension of deep Q-learning toward multi-agent learning led to remarkable results with regards to coordinated behavior and inter-agent communication. Based on this work, we may get new insights into swarm behavior and emergent phenomena. These novel ideas together with impressive empirical results make this a strong paper. Having said that, there are a few points I would like to address that may need improvement: 1. Clarity: Section 6.1 and 6.3 are very hard to understand. Even after multiple reads I was unable to understand how the color-digit MNIST game works. To assure reproducibility the authors should rewrite this section to make it more understandable. 2. Credit assignment: The underlying architecture for RIAL and DIAL is very complex (Section 6.3). On the other hand the point the authors make in this paper is very general: this is the first cooperative deep Q learning architecture where agents are able to learn communication protocols. It is unclear to me whether this is "a general concept that will work in a wide range of settings" or "a sophisticated engineering afford to make something work that didn't work before". While written very general in the Introduction, the experiments cater more toward the latter. To underline the generality there should be a simple toy example where the proposed method will work without the need for a sophisticated model architecture. Ideally this should be easily reproducible and already show cooperative behaviour. As it is now, it is hard to judge what limit the architecture imposes (e.g horizon length, non-linear expressiveness, possible divergence of q-learning,..,) and what drawbacks are due to a possible suboptimal algorithm. Minor: 209-211 double sentence

Confidence in this Review

2-Confident (read it all; understood it all reasonably well)


Reviewer 5

Summary

The paper considers problems involving multiple agents in an environment with a shared goal, and attempts to learn strategies to solve such collaborative tasks. The novelty of the work is to introduce differentiable real-valued message-passing between agents, so that their communication can be trained end-to-end in order to solve the collaborative tasks. The resulting Differential Inter-Agent Learning (DIAL) technique has been thoroughly evaluated on complex games like the switch riddle and MNIST digit recognition games to evaluate its utility and convergence to optimal policy. A good empirical study has been carried out to compare DIAL and RIAL in the presence and absence of parameter sharing and the study shows encouraging results for DIAL.

Qualitative Assessment

Though the paper contains a very thorough experimental evaluation of the suggested DIAL technique for multi-agent settings and the reviewer understands that it would have taken a lot of time and effort to set up and evaluate the experiments, the paper does not make a very novel contribution. It is clear that idea of shared memory and passing message gradients between agents would speed up learning and help to find the optimal policy faster, but this might not be a very natural way to do it. For instance, humans working in teams do not have shared memories. Also, for humans, messages from other humans are a part of their observation at each time step, rather than separate signals which are treated specially as messages and optimized differently than the rest of the observation. The idea of passing message gradients is certainly useful to have trainable message protocols while training a set of agents to perform a repetitive task, but doesn't offer much insight or useful interpretation as to how humans perform tasks in teams. Furthermore, experimental evaluation has been done for very small instances of the games involved, e.g. only for n = 3 and 4 for the switch riddle. It is probably due to the size of the policy space involved. But this is exactly what is needed from a good reinforcement learning algorithm - combatting huge state spaces with intelligent exploration instead of random exploration. Even while going from n=3 to n=4, there is significant performance degradation involved in the sense that DIAL (with parameter sharing) fails to find the optimal policy despite the increase in number of epochs from ~5000 to ~50000. This somewhat reflects the non-scalability of the algorithm for many agents. Moreover since the paper relies highly on experimental evaluation, it might be useful to try more challenging and bigger tasks to evaluate the performance and scalability of the DIAL technique. Apart from this, a language and grammar check can improve the presentation somewhat. Also avoid repetitions, for example in the last paragraph of section 5.2. Lastly, at some places relaxing the notations a bit can enhance readability. For instance, in the color-digit MNIST in section 6.3, the reward r(a) was quite hard to interpret. It is also not clear what a_2^a is, since it hasn't been described in that section (perhaps the authors meant u_2^a). EDIT AFTER REBUTTAL: The authors have clarified some of my raised concerns, regarding scalability of the algorithm and its contrast to how groups of humans accomplish such tasks. I am convinced with the response and consequently I am improving my ratings slightly to reflect the same.

Confidence in this Review

2-Confident (read it all; understood it all reasonably well)